# Associations between Body Image and Self-Perceived Physical Fitness in Future Spanish Teachers

**DOI:** 10.3390/children9060811

**Published:** 2022-05-31

**Authors:** Jorge Rojo-Ramos, Santiago Gómez-Paniagua, Jorge Carlos-Vivas, Sabina Barrios-Fernandez, Alejandro Vega-Muñoz, Carlos Mañanas-Iglesias, Nicolás Contreras-Barraza, José Carmelo Adsuar

**Affiliations:** 1Social Impact and Innovation in Health (InHEALTH) Research Group, Faculty of Sport Sciences, University of Extremadura, 10003 Cáceres, Spain; sabinabarrios@unex.es (S.B.-F.); cmaanasi@alumnos.unex.es (C.M.-I.); 2BioẼrgon Research Group, University of Extremadura, 10003 Cáceres, Spain; 3Promoting a Healthy Society Research Group (PHeSO), Faculty of Sport Sciences, University of Extremadura, 10003 Cáceres, Spain; jorgecv@unex.es (J.C.-V.); jadssal@unex.es (J.C.A.); 4Public Policy Observatory, Universidad Autónoma de Chile, Santiago 7500912, Chile; alejandro.vega@uautonoma.cl; 5Facultad de Economía y Negocios, Universidad Andres Bello, Viña del Mar 2531015, Chile; nicolas.contreras@unab.cl

**Keywords:** body image, physical fitness, university students, well-being, perceptions

## Abstract

Body image is a complex construct related to how each person perceives their own body and how they value it. Physical fitness and physical activity are factors that can influence the perception of a better or worse body image. This study aimed to identify the potential associations between body image and physical fitness self-perception in future Spanish teachers, analysing possible sex-related differences. A total of 278 Spanish university students answered the Multidimensional Body Self Relations Questionnaire and the International Fitness Scale, having an average age of 22 years, of which 40% were men and 60% were women. Nonparametric techniques (Spearman’s Rho test) were used as the data did not fit normality. The findings showed associations between body image and perceived physical fitness, confirming differences between the sexes. Correlations were found between the first three dimensions of the Multidimensional Body Self Relations and the International Fitness Scale, with sex-related differences being more significant in women than in men, and between the physical abilities self-assessed by the International Fitness Scale (except flexibility) and the dimensions of the Multidimensional Body Self Relations (except Dimension 4). Since body image influences well-being and conditions the time spent exercising, public health organisations and universities should design supports to improve master students’ body image through physical activity programmes, education and sex-specific individualised attention.

## 1. Introduction

Body image (BI) is a complex and multidimensional construct defined as the set of beliefs and thoughts a person has about their own body, resulting in a subjective picture, irrespective of how their body looks [1,2]. It comprises cognitive (thoughts and beliefs), affective (feelings), behavioural (actions) and perceptual (perceptions) components [3], influenced by culture, social pressure and media, among others [1]. Poor BI is commonly characterised by unhappiness with one’s appearance [4], affecting self-esteem, mood, competence, and social and occupational functioning. Moreover, body image distortions and body dissatisfaction can lead to unhealthy habits such as gaining weight or avoiding places where they may feel exposed [5,6], and even more dangerous behaviours such as self-starvation, steroid use or gym attendance abuse [7]. Moreover, negative BI is linked to low self-esteem [8], depression [9], social anxiety [10] and reduced sexual functioning [11]. It also leads to developing and maintaining body dysmorphic disorder and eating disorders [12,13]. Therefore, BI assessment and surveillance are essential across the lifespan, from childhood to adulthood, and are especially critical during adolescence [6,14,15]. Multiple dimensions of one’s body image may be implicated in the participation of physical activity and sport behaviour [16], since body image is relevant as a correlate, antecedent and consequence of physical activity behaviour [17].

Physical fitness (PF) is a person’s ability to perform activities of daily living with optimal performance, endurance and strength. It comprises five main categories: morphological, muscular, motor, cardiorespiratory and metabolic [18]. Physical activity (PA) is defined as any bodily movement produced by skeletal muscles requiring energy expenditure and related to different health benefits [19]. The associations between BI and PA have been studied since PA is considered a cause and effect of sporting behaviour [2,20]. Practising PA positively impacts physical perception and increases self-efficacy, confidence and BI [8]. Other authors emphasise that perceived improvements in physical capacities play an essential role [21] as individuals with positive BI are more likely to engage in PA [22]. Fitness training interventions or programs may also improve positive BI by encouraging individuals to focus more on their functionality and less on their physical appearance [23]. In addition, participating in potentially embodying activities where PA is involved but is not the primary goal of the activity, such as music performance [24] and acting [25], would promote positive BI through an appreciation of the body’s capabilities and the mind–body connection. In this sense, a positive BI may encourage people to adhere to PA programs [26]. However, other studies warn of the potential adverse effects of addressing PF, with increased preoccupations with weight, appearance and attractiveness [27]. In summary, there are two opposing approaches regarding the relationship between BI and PA [28]: those that claim that PA participation is related to a positive BI, with active subjects experiencing a more positive attitude towards their own body than sedentary subjects, and the benefits for their physical and mental health [29,30]; and, conversely, those who warn of potential adverse effects related to sports performance, the subject’s body perception and the possibility of suffering from ED, with this risk being correlated with the type of sport practised [31].

One of the most researched aspects is possible sex differences [32]. Women are more influenced by aesthetic body models, which emphasise their bodies’ physical attractiveness [33]. They are also more dissatisfied because they overestimate their body weight and try to be thinner [34]; thus, a strong association exists between body dissatisfaction and their body mass index [35]. By comparison, males’ BI can be strongly influenced by how they align with male stereotypes, so they try to be stronger [36], although they value their body’s physical capabilities [37] and are more satisfied with their physical appearance than women [38].

Young adults are a very important group in terms of being overweight or underweight [29], and experiencing mostly serious problems with body dissatisfaction [39]. The connection between body satisfaction and physical activity remains a topic of ongoing discussion; bad body image can be a motivator or a deterrent to physical activity, and can lead to social physique anxiety based on real or imagined negative physical judgments [40,41]. Additionally, the COVID-19 pandemic has significantly impacted food consumption, sedentary behaviour and PA levels [42], especially among university students, worsening their sedentary behaviours and diet [43,44]. In terms of age, the relationship between university students’ age and body image perceptions appears to have received little attention in the literature, possibly due to the narrow age bands observed in typical college student populations [45]. The insistence of public institutions that individuals maintain regular daily PA [46] has produced adverse consequences during the lockdown, increasing the gain in weight, anxiety, and depression [44,47]. Due to the pandemic and post-pandemic context and socioeconomic status, university students may be at risk of experiencing mental health and social issues [48,49]. Therefore, this study examined the associations between BI and PF self-perception in future teachers of the autonomous community of Extremadura (Spain), in addition to analysing potential sex differences. Thus, we asked the following research question: Are there significant associations between the dimensions of the Multidimensional Body Self Relations Questionnaire (MBSRQ) and the International Fitness Scale (IFIS) in university students in the region of Extremadura?

## 2. Materials and Methods

### 2.1. Study Design and Procedure

This was a cross-sectional observational study. Six professors from the Department of Musical, Plastic and Corporal Expression of the University of Extremadura shared information about the study goals through their subjects’ virtual classrooms with their students. They provided access to the informed consent form and a URL link to the sociodemographic survey, the MBSQR [50] and the International Fitness Scale [51]. A time response of 10 min was estimated. Data were collected from February to March 2022.

All data were collected anonymously and kept private. The study was performed according to the Declaration of Helsinki guidelines and, following the Regulation 2016/679 of the European Parliament [52], no bioethics committee approval was required because all responses and information received were anonymous.

### 2.2. Participants

The sample consisted of 278 future teachers (Master’s students) from the public University in Extremadura (Table 1), the public centre of this region of Spain, representing 63% of the total number of Master’s students in the different training modalities. The median age was 21 years (interquartile range = 4). Participants were selected using a convenience sampling method [53].

### 2.3. Instruments

The sociodemographic survey questions related to sex, age, and degree completed.

The Spanish version of the Multidimensional Body Self Relations Questionnaire (MBSRQ) [50] was used, as translated, culturally adapted and validated by del Cid and colleagues [54]. Similarly, its validity was tested by comparing the responses between university students and surgery patients, and in the pre- and post-surgery of the aforementioned patients. The Spanish version is composed of 45 items grouped into four factors: (a) Dimension 1, “Subjective importance of corporeality”, joins thirty items about concerns regarding physical appearance, activities to maintain physical shape, considerations about weight and dieting, concerns about health and sickness, and perceptions of body parts; (b) Dimension 2, “Behaviors related to preserving physical shape”, consists of seven items that assess self-perceived physical shape and physical form orientation; (c) Dimension 3, “Self-assessed physical attractiveness”, comprises three items about self-assessing physical attractiveness; and (d) Dimension 4, “Care for physical appearance”, comprises five items assessing physical appearance concerns. The indirect items were reversed for statistical analysis. Responses use a Likert scale (1–5), with 1 being “strongly disagree”, 2 “strongly disagree”, 3 “indifferent”, 4 “strongly agree”, 5 “strongly agree”. In addition, authors reported an overall Cronbach’s alpha value of 0.88 [54].

In addition, the Spanish version of the International Fitness Scale (IFIS) adapted to the young adult population was applied. The instrument was first translated and adapted to Spanish by Ortega et al. [55] in the youth population. More recently, it was validated by Español-Moya and Ramírez-Vélez in the university population [56], reporting an overall Cronbach’s alpha value of 0.82.

This instrument comprises five items to ascertain general PF self-perception, cardiorespiratory fitness, muscular strength, flexibility, and speed–agility. It is answered using a scale of 1: very bad, 2: bad, 3: acceptable, 4: good, and 5: very good.

### 2.4. Statistical Analysis

The SPSS statistical software version 23 for MAC (IBM SPSS, Chicago, IL, USA) was used to process the data. Firstly, the Kolmogorov–Smirnov test was employed to check normality in the continuous variables’ data distribution; as this assumption was not met, nonparametric statistical tests were chosen. Spearman’s Rho test was used to analyse the relationship between the MBSRQ and IFIS dimensions. Thus, to interpret correlation coefficients, thresholds proposed by Mondragón-Barrera [57] were followed: from 0.01 to 0.10 (low correlation), from 0.11 to 0.50 (medium correlation), from 0.51 to 0.75 (strong correlation), from 0.76 to 0.90 (very high correlation) and from 0.91 to 1.00 (perfect correlation). Cronbach’s alpha was used to analyse the reliability of each instrument; to interpret the values, Nunally and Bernstein criteria were chosen [58]: <0.70 (low), 0.71 to 0.90 (satisfactory) and >0.91 (excellent).

## 3. Results

Table 2 and Table 3 report the descriptive statistics of the dimensions of the MBSRQ and IFIS questionnaires, respectively.

Table 4 shows the correlations between the MBSRQ dimensions and the IFIS, and analysed according to sex. There were positive, considerable (>0.51) and significant (*p* < 0.001) associations between Dimensions 1 “Subjective importance of corporeality” and 2 “Behaviors aimed at maintaining PF” and the IFIS. Moreover, a positive, average (0.46) and significant (<0.001) association was found between Dimension 3, “Self-rated physical attractiveness”, and the IFIS. There were stronger associations in men compared to women.

Table 5 shows the associations between the physical abilities self-assessed by the IFIS and the MBSRQ dimensions. General PF has a positive, considerable, and significant association with Dimension 1, “Subjective importance of corporeality”, and a positive, moderate and significant association with Dimensions 2, “Behaviors aimed at maintaining PF”, and 3, “Self-assessed physical attractiveness”. Cardio-respiratory fitness has a positive, considerable, and significant association with Dimensions 1 and 2 and a positive, moderate, and significant association with Dimension 3. There is a positive, significant association between the muscular strength variable with Dimension 2 and positive, moderate, and significant association with Dimensions 1 and 3. Concerning the variable speed/agility, a positive, moderate, and significant association was found with Dimensions 1, 2 and 3. However, no associations were found between the flexibility and the MBSRQ dimensions, nor between any of the self-assessed physical capacities and Dimension 4, “Care of physical appearance”.

Finally, the reliability values were satisfactory for the MBSRQ dimensions using the Cronbach’s Alpha (Dimension 1 = 0.79; Dimension 2 = 0.82; Dimension 3 = 0.82; Dimension 4 = 0.83) and for the IFIS (0.80).

## 4. Discussion

### 4.1. Theoretical Implications

This study examined the associations between BI and perceived PF among future teachers in the region of Extremadura (Spain), checking for potential sex-related differences. We found correlations between the dimensions of the MBSRQ (BI): Subjective importance of corporeality, behaviours related to preserving physical shape and self-assessed physical attractiveness; and the IFIS (PF), with sex-related differences being more significant in females than males. The associations between the IFIS self-assessed physical abilities and the MBSRQ dimensions were also reported. Overall FP, cardiorespiratory fitness, muscle strength, and speed/agility were associated with Dimensions 1, “Subjective importance of embodiment”, 2, “Behaviors aimed at maintaining FP”, and 3, “Self-rated physical attractiveness”. However, no associations were found with either flexibility (IFIS) or the MBSRQ Dimension 4 “Care of physical appearance”.

Considering the MBSRQ and the IFIS outcomes, Tylka and Wood [27] noted the strong relationship between body appearance and physical condition. Generally, previous studies showed that women tend to give more importance to their appearance [59], usually focusing on their body linearity, in contrast to men, who often focus on muscularity [60]. However, and referring to the correlations found in the first dimensions of the MBSRQ, the most significant relationship was found between the importance given to physical appearance by men, which may be due to the change in their stream of thought in the transition from high school to college [61]. Regarding the link between the behaviours to keep the physical shape (Dimension 2) and the IFIS punctuation, previous studies have shown that those individuals with higher PA levels pay particular attention to activities to maintain physical appearance and well-being [16]. In most research, females show greater concern for maintaining a good physical appearance [62], although our results may be conditioned by the greater physical condition of the male sex in university, as males a wider range of physical activities in this age group [63]. Focusing on the association between self-perceived attractiveness and PF, the scientific literature notes that people with higher PF show less body dissatisfaction [64,65] and consider themselves more attractive. These studies also show that women tend to be more dissatisfied with their bodies [66,67], which is in agreement with the results of the study, as men seem to attach less importance to their appearance.

The relationship between BI and PF components is a field to explore since most research focuses on improving BI through PA interventions [17], or observing how interventions affect BI in specific populations [68,69]. Thus, better PF is positively related to all the MBSRQ dimensions, except for flexibility. The fitter an individual, the more strategies they adopt to stay active and the greater attractiveness they self-perceive [70]. Otherwise, this relationship is reduced in terms of physical appearance care because, in recent years, there has been a change in the approach to physical training, toward seeking to be fit rather than thin [71], especially in women [72]. In addition, some projects reported strong associations between BI and aerobic exercise for up to eight years after the initial intervention [73]. Similarly, strength training appears to be positively associated with BI [74,75], significantly improving all their dimensions [76].

### 4.2. Practical Implications

BI influences social, emotional, physical and psychological well-being, conditioning the amount of time spent exercising and increasing the risk of physical inactivity and sedentary lifestyles, and, therefore, the risk of numerous chronic diseases and mortality [77,78]. Adolescents and university students present a high risk of suffering mental issues due to the sustained stress in their daily life [79,80]. In addition, their social and economic situation can be either protective or risk factors for proper BI, PA level and PF [1,81].

As a result, public health organisations and universities should design supports to improve future teachers’ BI by providing PA programs, education and individualised psychological attention [82,83]. Moreover, interventions must have a sex-related perspective due to the differentiated sex-related characteristics [28], focused on the acquisition of PA habits and the enjoyment of practicing PA, especially in women, and considering increased risk for females and little attention dedicated to males [84]. In this sense, and according to the results, women should be able to enjoy exercise programs and activities that not only allow them to improve their physical condition but also their appearance. In the case of the male gender, physical activity should be oriented to improve their own perception of certain body parts, which will lead them to improve their self-image.

### 4.3. Limitations and Future Lines

This study has several limitations. As the total sample originates from the Spanish Region of Extremadura, the sociocultural context may influence the results. In addition, convenience sampling was undertaken, so there was no randomisation. Additionally, this type of study design does not allow the establishment of cause–effect relationships.

Different future lines of research can be considered, such as a multicentre study to determine whether cultural and/or socioeconomic differences may influence the results between various regions of Spain. Moreover, it would be interesting to compare the BI and PA among students from different knowledge areas (sciences, health sciences, education, social sciences, humanities and engineering) or with students from the Baccalaureate or Professional Training programmes.

## 5. Conclusions

The findings of this study seem to indicate certain associations between BI and perceived PF among future teachers in Extremadura, confirming differences between the sexes, although further in-depth studies are obviously needed to confirm this. Correlations were found between some dimensions of the MBSRQ (BI) and the IFIS (PF), with sex-related differences being more significant in women than in men, and between the physical abilities self-assessed by the IFIS (except flexibility) and the dimensions of the MBSRQ (except Dimension 4).

## Figures and Tables

**Table 1 children-09-00811-t001:** Frequency distribution of the sample (N = 278).

Variable	Categories	N/M	%
Gender	Male	112	59.7
	Female	166	40.3
Age	Under 20	31	11.2
	Between 20 and 30	236	84.9
	Over 30	11	3.9
Degree	Teacher training	240	86.3
	Education	25	9
	Sports Sciences	13	4.7

**Table 2 children-09-00811-t002:** Descriptive statistics by sex of the MBSRQ dimensions.

MBSRQ Dimensions	Gender		
	Total	Male	Female
	M_e_ (IQR)	M_e_ (IQR)	M_e_(IQR)
(1) Subjective importance of corporeality	3 (1)	3 (1)	3 (0)
(2) Behaviors related to preserving physical shape	3 (1)	4 (1)	3 (1)
(3) Self-assessed physical attractiveness	4 (1)	4 (1)	3 (1)
(4) Care for physical appearance	4 (1)	4 (1)	4 (1)

Note: M_e_ = median value; IQR = interquartile range. Each score obtained is based on a Likert scale (1–5): 1 is “Strongly disagree” and 5 “Strongly agree”.

**Table 3 children-09-00811-t003:** Descriptive statistics by sex of the IFIS dimensions.

IFIS Dimensions	Gender		
	Total	Male	Female
	M_e_ (IQR)	M_e_ (IQR)	M_e_(IQR)
(1) General physical condition	3 (1)	4 (1)	3 (1)
(2) Cardiorespiratory fitness	3 (2)	4 (2)	3 (1)
(3) Muscular strength	3 (1)	4 (1)	3 (1)
(4) Speed-agility	4 (1)	4 (0)	3 (1)
(5) Flexibility	3 (2)	3 (2)	3 (2)

Note: M_e_ = median value; IQR = interquartile range. Each score obtained is based on a Likert scale (1–5): 1 is “Strongly disagree” and 5 “Strongly agree”.

**Table 4 children-09-00811-t004:** Correlations between the MBSRQ dimensions and the IFIS questionnaire.

MBSRQ Dimensions	IFIS *ρ (p)*	IFIS *ρ (p)*	
		Male	Female
(1) Subjective importance of corporeality	0.57 (<0.001)	0.59 (<0.001)	0.45 (<0.001)
(2) Behaviors related to preserving physical shape	0.57 (<0.001)	0.56 (<0.001)	0.34 (<0.001)
(3) Self-assessed physical attractiveness	0.46 (<0.001)	0.44 (<0.001)	0.40 (<0.001)
(4) Care for physical appearance	0.06 (0.272)	0.18 (0.048)	−0.10 (0.194)

MBSRQ: The Multidimensional Body Self Relations Questionnaire; IFIS: International Fitness Scale. Spearman’s Rho test was used.

**Table 5 children-09-00811-t005:** Correlations between MBSRQ dimensions and the IFIS questionnaire.

	IFIS Questionnaire				
MBSRQ Dimensions	General Physical Condition *ρ (p)*	Cardiorespiratory Fitness *ρ (p)*	Muscular Strength *ρ (p)*	Speed-Agility *ρ (p)*	Flexibility *ρ (p)*
Subjective importance of corporeality	0.59 (<0.001)	0.60 (<0.001)	0.36 (<0.001)	0.34 (<0.001)	0.08 (0.18)
Behaviors related to preserving physical shape	0.49 (<0.001)	0.54 (<0.001)	0.53 (0.001)	0.32 (<0.001)	0.06 (0.312)
Self-assessed physical attractiveness	0.46 (<0.001)	0.43 (<0.001)	0.22 (<0.001)	0.37 (<0.001)	0.14 (0.01)
Care for physical appearance	0.07 (0.21)	0.01 (0.75)	0.04 (0.445)	0.01 (0.93)	0.10 (0.09)

MBSRQ: The Multidimensional Body Self Relations Questionnaire; IFIS: International Fitness Scale. Spearman’s Rho test was used.

## Data Availability

The datasets used during the current study are available from the corresponding author on reasonable request.

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
