# Peer review of "Associations between Body Image and Self-Perceived Physical Fitness in Future Spanish Teachers"

_children, 2022, doi:10.3390/children9060811_

Round 1

Reviewer 1 Report

Thank you very much for the opportunity to review this article.

The content is interesting. I have no comments, I am waiting for further research on this topic.

Author Response

thank you very much for your encouragement and interest in the article.

Reviewer 2 Report

The topic of manuscript is not novel and there is a substantial amount of research reporting associations between more positive body image and better perceived physical fitness. The novelty of research is low, and this research does not help to better understand the associations between body image and physical activity. The rationale for the study is not well established.

Further, the sample is too low, and the conclusions are overstated. It could not be concluded that the tendencies found is such a small sample might be reflected in all Spain students (p. 6, lines 235-239).

Next my concern is ethical permission. I could not agree that this study might be implemented without the permissions of ethic committees of university or other levels (p. 2, lines 94-96).

The names of the subscales are confusing, please address the names of the subscales of MBSRQ scale in original version. Cronbach alphas of all subscales must be presented.

Statistics of the manuscript is weak and incomplete, too small sample might be the reason since it is impossible to have enough statistical power having such a small sample.

I suggest authors to collect more data, to consider the ethical permissions and to rewrite the article.  

Author Response

The topic of manuscript is not novel and there is a substantial amount of research reporting associations between more positive body image and better perceived physical fitness. The novelty of research is low, and this research does not help to better understand the associations between body image and physical activity. The rationale for the study is not well established.

  • Response: We understand that at first glance the study does not seem novel, because as you mention there is a large literature that positively associates both concepts. However, the study is carried out in university students who have already been defined as a population with a high risk of suffering physical and mental problems due to their vital, economic, social and psychological situation. Therefore, it is necessary to establish such relationships between both aspects in this population in order to reverse the bad trend that has been developing during the last decade in Extremadura.

Further, the sample is too low, and the conclusions are overstated. It could not be concluded that the tendencies found is such a small sample might be reflected in all Spain students (p. 6, lines 235-239).

  • Response: We would like to apologize for the confusion, the study focuses on university students in the region of Extremadura. This fact has already been modified in the objective and, as you rightly point out, also in the conclusions in order to make the reader understand the dimension of the study.

Next my concern is ethical permission. I could not agree that this study might be implemented without the permissions of ethic committees of university or other levels (p. 2, lines 94-96).

  • Response: we understand the confusion that this factor of bioethics committees can generate, but in the words of the Regulation (EU) 2016/679 of the European Parliament and of the Council of 27 April 2016: “the data protection principles should not apply to anonymous information, i.e. information that does not relate to an identified or identifiable natural person, nor to data rendered anonymous in such a way that the data subject is not or no longer identifiable. Consequently, this Regulation does not affect the processing of such anonymous information, including for statistical or research purposes”. So we understand is not necessary the permission of ethic committee.

The names of the subscales are confusing, please address the names of the subscales of MBSRQ scale in original version. Cronbach alphas of all subscales must be presented.

  • Response: The questionnaire’s dimensions names are from the MBSRQ validated in the Spanish population (citation 46), therefore the names are a literal translation from Spanish. Also, Cronbach’s Alpha from all subscales are presented after Table 3.

Statistics of the manuscript is weak and incomplete, too small sample might be the reason since it is impossible to have enough statistical power having such a small sample.

  • Response: The sample is composed of 278 students from the University of Extremadura, and therefore the transferability of the results could only be applied to students from this region. This question has been corrected both in the introduction and in the conclusions to make it understandable.

I suggest authors to collect more data, to consider the ethical permissions and to rewrite the article.  

Reviewer 3 Report

Dear authors,

Thank you very much for your valuable work and submitted a very interesting manuscript. Please, find my review and suggestions in the attachment. I hope that suggestion with help and support you with your publication.

Best regards and all the best,
Reviewer

Author Response

I recommend avoiding abbreviations in the abstract. Please remove all abbreviations in the

brackets and introduce them in the Introduction or later (when they first appear from the

Introduction onwards ).

  • Response: the abbreviations question has been modified following your recommendations.

It would be beneficial for the understanding of the study if you would indicate the average

age and gender structure of your participants already in the abstract.

  • Response: Average age and gender structure are now mentioned in the abstract.

Please indicate which non-parametric tests were used.

  • Response: the Spearman’s Rho test were used, since it was not mentioned, it has now been included in the summary.

Introduction

Good written introductory paragraph. I would like to suggest that you look at the phenomena

under study from the point of view of the selected age group of the population (Spanish

students). Please justify why the relationships between BI and PF need to be studied in

this age group in particular. This was briefly mentioned in the last paragraph of the

introduction, but should be justified more strongly in the literature review.

  • Response: the relationships between BI and PF has been amplified and justified in the last introduction’s paragraph to ease the understanding.

Line 49, 2nd paragraph: It would be beneficial to insert a linking sentence between the first

paragraph (Body Image) and the second paragraph (Physical Fitness). How these two

phenomena are related or why PF it is important to study them in a context with BI.

  • Response: the relationship between Bi and PF has been explained as a linking sentence in the final part of the first paragraph following your recommendation.

It would be greatly appreciated if you could clarify the research questions at the end of the

introduction.

  • Response: the research question is now included to clarify the main objective of the manuscript.

Materials and Methods

2.2 Participants: Please define the type of university selected (private, public).

  • Response: the type of university has been defined in line 115.

2.1 (Ethical considerations) and 2.3 (Procedure) are linked. Please summarise them in one

section.

  • Response: subsections 2.1 and 2.3 have been summarized in one following your indication.

2.4: The instruments are well described. However, information on their validity and reliability

is missing. Please also describe how the measurement tools have been translated and

tested for use in Spanish.

  • Response: The methods for translation as well as reliability/validity indexes has been included in the descriptions of both questionnaires.

Line 124: The International Fitness Scale (IFIS) is written in bold.

  • Response: The terms written in bold has been corrected.

2.5, lines 136-137: "considerable correlation" - Please consider using a different term, e.g.

"strong correlation".

  • Response: the expression “considerable correlation” has been substituted by other term following your indication.

Results

Lines 168-170: The reliability value should be moved to section 2.4 Instruments.

  • Response: these reliability values belong to the statistical analysis of the manuscript, therefore we consider that they should be included in the results section. Reliability values from previous studies that form the research basis of these scales are mentioned in the instruments subsection.

Table 2: Following the study objective "...this study examines the correlations between BI and

PF self-perception in Spanish undergraduates, analysing potential sex differences", I

would like to suggest that Table 2 (Correlations between MBSRQ dimensions and the IFIS

questionnaire) should include the separate analysis for women and men (e.g. Tables 3 and

4) in addition to the analysis of the whole sample.

  • Response: that is a really good point. However, the objective of the study is to relate both questionnaire in order to obtain if there is any strong association between their dimensions. Obviously, gender differences are well described in both fields by previous literature, BI and PF. Table 2 relate both questionnaires attending gender differences, grouping MBRSQ in dimensions and referring IFIS at a general way. Analyzing gender differences in Table 3 is redundant, as the IFIS scale only have 5 items, but looking for associations between them and MBRSQ dimensions will serve as a starting point for future studies where we will look inside gender differences observing IFIS’ items separately.

Please consider including in the results the main descriptive statistics for the MBSRQ

dimensions and the IFIS and discuss them in the discussion.

  • Response: the results of the main descriptive statistics for both questionnaires has been included in table 2 and 3.

Discussion

The discussion still needs to be elaborated.

4.1: First paragraph: Perhaps the numbers of the dimensions are not so important for the

discussion. The focus should be on the findings (content).

  • Response: First paragraph is a brief description of the correlations founded in the study to introduce the discussion. Following your next indication, we will develop the second paragraph to present the main findings and discuss them.

4.1: Second paragraph seems like a literature review. Section 4.1 should present the main

findings of the manuscript and discuss them in the context of other studies.

  • Response: the second paragraph discuss the findings (correlations) one by one, however, it is true that remains unclear for the change of the terms shown in the tables- Thus, this second paragraph has been extended and developed in a clearly way. understand that there is little literature relating body image in general and physical fitness levels, even less so in university populations. Therefore, the formulations of previous studies should be carefully extracted.

4.2: Perhaps you could add some of your original practical implications and not only literature

review. For example, how should gender characteristics be taken into account in

interventions based on the results of your study?

  • Response: Practical implications subsection has been developed following your recommendations.

Conclusion

"The findings of this study indicate associations between BI and perceived PF among 235

university students in Spain, confirming differences between the sexes." - Please take into

account in this conclusion that the convenience sample does not allow for generalisation.

  • Response: the sentence has been corrected due to the procedure developed in the study.

The number of dimensions is not so relevant to the conclusion. The focus should be on the

content.

  • Response: This question has been corrected.

References

Please, check carefully a reference number 71 – are the capital letter in the title needed?

  • Response: the title of this reference is now well written.

Author Contributions

Please consider defining the role of each co-author in more detail. Entrusting all the tasks of

the manuscript to all the authors does not seem convincing and casts doubt on the

authors' legitimate contributions.

  • Response: Authors’ contributions has been included.

Round 2

Reviewer 2 Report

Authors addressed some my comments, however, the sample size is too low and this compromises the research. Novelty of the research is not clear and conclusions driven from this pilot study are doubtful.

Author Response

Authors addressed some my comments, however, the sample size is too low and this compromises the research. Novelty of the research is not clear and conclusions driven from this pilot study are doubtful.

  • Results: the sample represents 63% of the master's students in the entire region, therefore the sample cannot be considered low (the population focus of analysis has changed). The novelty of the study lies in the analysis of the relationships between self-perceptions of body image and physical condition in future teachers in the Spanish region of Extremadura (never before evaluated). The conclusions refer to statistically significant differences and correlations, without giving rise to total statements due to the procedures used and reiterating the need for a more exhaustive study.

Reviewer 3 Report

Dear authors,

Congratulations on the very precisely elaborated improvements of the manuscript! Thank you for carefully considering all suggestions in the review. In the process of the final stage of publication, I would like to suggest some additional minor revisions:

The percentage in the abstract could be written with the symbol: %.

Introduction, Line 105: please, define the abbreviations MBSRQ and IFIS in the introduction. This is the first time two abbreviations appear in the text, and they need to be clarified.

2.1 Procedure: I would like to suggest starting the section with the sentence about research design “This is a cross-sectional observational study.” Maybe the title “Study design and procedure” would better fit this part. Ethical considerations may stay in this section or be presented in a separate section within Materials and Methods.

Discussion, 2nd paragraph (Lines 229-248): p results are not clear. For example, “in most research, females (p = 0.34) show greater…” – it is unclear how p was calculated as findings of more studies were mentioned. Please, consider and be more precise in describing your findings and the findings of other studies. Your study findings need to be recognised and compared with similar studies. This comparison is the weakest part of the manuscript and should be more convincing.

Discussion, 4.2: Perhaps you could add some of your original practical implications. For example, how should gender characteristics be considered in interventions based on your findings?

Thank you very much for your valuable work!

Author Response

Congratulations on the very precisely elaborated improvements of the manuscript! Thank you for carefully considering all suggestions in the review. In the process of the final stage of publication, I would like to suggest some additional minor revisions:

The percentage in the abstract could be written with the symbol: %.

  • Response: Corrected

Introduction, Line 105: please, define the abbreviations MBSRQ and IFIS in the introduction. This is the first time two abbreviations appear in the text, and they need to be clarified.

  • Response: Corrected

2.1 Procedure: I would like to suggest starting the section with the sentence about research design “This is a cross-sectional observational study.” Maybe the title “Study design and procedure” would better fit this part. Ethical considerations may stay in this section or be presented in a separate section within Materials and Methods.

  • Response: all suggestions have been included. Ethical considerations will stay in this subsection.

Discussion, 2nd paragraph (Lines 229-248): p results are not clear. For example, “in most research, females (p = 0.34) show greater…” – it is unclear how p was calculated as findings of more studies were mentioned. Please, consider and be more precise in describing your findings and the findings of other studies. Your study findings need to be recognised and compared with similar studies. This comparison is the weakest part of the manuscript and should be more convincing.

  • Response: some parts of the discussion have been rewritten, facilitating the reader's understanding.

Discussion, 4.2: Perhaps you could add some of your original practical implications. For example, how should gender characteristics be considered in interventions based on your findings?

  • Response: following your suggestion, practical implications have been developed.

Thank you very much for your valuable work!